# Inhibition of Osteoclast Differentiation by Carotenoid Derivatives through Inhibition of the NF-κB Pathway

**DOI:** 10.3390/antiox9111167

**Published:** 2020-11-23

**Authors:** Shlomit Odes-Barth, Marina Khanin, Karin Linnewiel-Hermoni, Yifat Miller, Karina Abramov, Joseph Levy, Yoav Sharoni

**Affiliations:** 1Clinical Biochemistry and Pharmacology, Faculty of Health Sciences, Ben-Gurion University of the Negev, Beer-Sheva 84105, Israel; shlomitbarth@post.bgu.ac.il (S.O.-B.); hanin@bgu.ac.il (M.K.); Karin.Hermoni@lycored.com (K.L.-H.); lyossi@bgu.ac.il (J.L.); 2Department of Chemistry, Ben-Gurion University of the Negev, Beer-Sheva 84105, Israel; ymiller@bgu.ac.il (Y.M.); karinaab@post.bgu.ac.il (K.A.); 3Ilse Katz Institute for Nanoscale Science and Technology, Ben-Gurion University of the Negev, Beer-Sheva 84105, Israel

**Keywords:** apo-carotenals, lycopene, polyphenols, bone, osteoclasts, NFκB, synergy

## Abstract

The bone protective effects of carotenoids have been demonstrated in several studies, and the inhibition of RANKL-induced osteoclast differentiation by lycopene has also been demonstrated. We previously reported that carotenoid oxidation products are the active mediators in the activation of the transcription factor Nrf2 and the inhibition of the NF-κB transcription system by carotenoids. Here, we demonstrate that lycopene oxidation products are more potent than intact lycopene in inhibiting osteoclast differentiation. We analyzed the structure–activity relationship of a series of dialdehyde carotenoid derivatives (diapocarotene-dials) in inhibiting osteoclastogenesis. We found that the degree of inhibition depends on the electron density of the carbon atom that determines the reactivity of the conjugated double bond in reactions such as Michael addition to thiol groups in proteins. Moreover, the carotenoid derivatives attenuated the NF-κB signal through inhibition of IκB phosphorylation and NF-κB translocation to the nucleus. In addition, we show a synergistic inhibition of osteoclast differentiation by combinations of an active carotenoid derivative with the polyphenols curcumin and carnosic acid with combination index (CI) values < 1. Our findings suggest that carotenoid derivatives inhibit osteoclast differentiation, partially by inhibiting the NF-κB pathway. In addition, carotenoid derivatives can synergistically inhibit osteoclast differentiation with curcumin and carnosic acid.

## 1. Introduction

Several epidemiological studies imply that fruit and vegetable consumption decreases morbidity and has a beneficial effect on bone health [1,2,3]. Carotenoids, a major group of micronutrients in a fruit and vegetable-rich diet, are fat soluble and pigmented phytochemicals produced by bacteria, fungi, algae, and plants [4]. From the more than 600 natural carotenoids that have been identified, nearly 50 are consumed by humans [5], whereas about 20 appear in human tissues and blood [6]. β-carotene, lycopene, and lutein compose the major plasma carotenoids [7]. Lycopene is derived mainly from tomatoes and tomato products, and its content in tomatoes is 0.7–20 mg/100 g wet weight [8]. The sources of other carotenoids are more diverse; for example, β-carotene is rich in orange-yellow vegetables and fruits, but it is also found in leafy vegetables. Humans appear to absorb carotenoids in a relatively non-specific fashion and, thus, their plasma and tissue concentrations reflect their individual dietary habits [7]. The relative abundance of each of the five major carotenoids in the diet are similar to their distribution in plasma.

The role of carotenoids has been investigated in epidemiological and interventional studies. Lycopene supplementation to postmenopausal women for four months significantly decreased oxidative stress parameters and the bone resorption marker n-telopeptide of type I collagen. This was accompanied by a significant increase in serum lycopene. Most adult bone diseases are due to excess osteoclastic activity, which results in an imbalance in bone remodeling which favors resorption by osteoclasts over building by osteoblasts [9]. Animal and cellular studies on the role of fruits and dietary phytochemicals in bone protection were reviewed by Shen et al. [10]. An in-vivo study showed that a supplement containing tomatoes improved bone health in ovariectomized osteoporotic rats [11]. The effect of carotenoids on bone has also been studied in cell culture. Rao et al. showed that the carotenoid lycopene stimulates osteoblast cell proliferation and alkaline phosphatase activity in SaOS-2 cells, inhibiting osteoclast formation and mineral resorption mediated by reactive oxygen species in cells from rat bone marrow [12,13]. Costa-Rodrigues et al. studied the effects of lycopene on differentiation and function in human osteoclasts and osteoblasts. They found that lycopene decreased osteoclast differentiation and resorbing activity, and increased osteoblast proliferation and differentiation [14]. Using signaling inhibitors, they tried to identify the pathways involved in lycopene action but were unable to show an effect on NF-κB in osteoclasts even though such an effect was found in osteoblasts.

Identification of the osteoclastogenesis inducer, RANKL, expressed mainly in osteoblasts; its cognate receptor, RANK, expressed on osteoclast progenitors; and its decoy receptor osteoprotegerin has contributed to understanding of the molecular mechanisms of osteoclast differentiation and activity [15]. One of the early molecular events induced by RANK is NF-κB activation [16,17]. In non-stimulated cells, NF-κB proteins are found in the cytoplasm, but enter the nucleus upon cell stimulation. The NF-κB pathway is composed of two distinct pathways: the canonical and the alternative. Both are shown to be essential in osteoclastogenesis [17,18,19,20]. NF-κB transcription factor activity is the hallmark of inflammation. In this respect, the role of lycopene as an anti-inflammatory agent was studied. Joo et al. [21] demonstrated that tomato lycopene extract inhibits NF-κB signaling, leading to reduced-lipopolysaccharide-induced pro-inflammatory gene expression in rat small intestinal epithelial cells. A similar anti-inflammatory effect of lycopene was shown in lipopolysaccharide-induced peritoneal macrophages [22]. These findings were supported by a study showing that lycopene regulates cigarette smoke-driven inflammation by inhibition of macrophage NF-κB activity [23]. However, whether inhibition of NF-κB signaling is involved in lycopene’s effect in osteoclasts is not yet known.

In several types of cell including bone osteoblasts, we have previously shown that carotenoid oxidation products, and not the intact carotenoid, stimulate the electrophile/antioxidant response element (ARE/Nrf2) transcription system [24] and inhibit the NF-κB transcription system [25]. Similar opposing effects on these two transcription systems were obtained with synthetic dialdehyde carotenoid derivatives (diapocarotene-dials), which can be formed by spontaneous oxidation [26] or after chemical [27] or enzymatic [28] catalyzed oxidation of various carotenoids. Although such diapocarotene-dials have not been identified in human or animal samples, mono-apocarotenals, that have similar, but lower activities [24], have been documented in raw tomatoes [29]. The synthetic diapocarotene-dials also inhibited estrogen signaling in breast cancer cells but did not inhibit and even stimulated it in osteoblast bone cells [30]. In addition, we demonstrated that the activity of individual diapocarotene-dials in inducing the ARE/Nrf2 transcription system and inhibiting the NF-κB transcription system depends on the reactivity of the conjugated double bond in reactions such as Michael addition. This reactivity is determined by the electron density around the reactive carbon atoms (the fourth atom from each side of the molecule; see Table 1) [25]. We hypothesized that oxidized derivatives of lycopene and other carotenoids also act as the active mediators in inhibiting osteoclast differentiation.

The aim of the current work was to determine if intact lycopene or its oxidized derivatives inhibit RANKL-induced osteoclast differentiation in RAW264.7 osteoclast progenitor cells. In addition, we determined the relative inhibition of osteoclast differentiation by various diapocarotene-dials in order to evaluate if the structure–activity relationship is similar to that of NF-κB inhibition [25]. After establishing this similarity, we aimed to verify if oxidized lycopene and the carotenoid derivatives interfere in the NF-κB pathway.

It is well accepted that the health benefits of a fruit and vegetable-based diet reside, at least in part, in additive or synergistic activities of their phytonutrients. We hypothesized that this is true also for the inhibition of osteoclast differentiation; thus, another aim of this study was to look for synergy between carotenoid derivatives and other phytonutrients. To check this possibility, we compared the inhibition of osteoclastogenesis by a carotenoid derivative alone to its combination with phytochemicals belonging to the large family of polyphenols, several of which are known to have beneficial effects on bone health. Polyphenols are present in many foods of plant origin and are characterized by having one or several phenolic groups in their chemical structure. There are over 500 different polyphenols in foods, and the mean intake of all of them is about 1 g per day, which is split between many specific polyphenols [31]. From the various polyphenols, we selected two which affect osteoclasts—curcumin [32,33] and carnosic acid [34,35]—and studied their cooperativity with carotenoid derivatives in inhibiting osteoclast activity.

## 2. Materials and Methods 

### 2.1. Materials

Crystalline lycopene preparations, purified from tomato extract (>97%), were supplied by Lycored Ltd. (Beer Sheva, Israel). Tetrahydrofuran (THF), containing 0.025% butylated hydroxytoluene (BHT) as an antioxidant, was purchased from Aldrich (Milwaukee, WI, USA). fetal calf serum (FCS), sodium pyruvate, and Ca^2+^/Mg^2+^-free PBS were purchased from Biological Industries (Beth Haemek, Israel). DMEM medium was purchased from Gibco (Grand Island, NY, USA). α-MEM medium, Dimethyl sulfoxide (DMSO), P-nitrophenyl phosphate and acid phosphatase leukocyte kit (387A) were purchased from Sigma Chemicals. Curcumin was purchased from Cayman Chemicals (Ann Arbor, MI, USA). Carnosic acid was purchased from Alexis Biochemicals (Läufenfingen, Switzerland).

### 2.2. Ethanolic Extract of Lycopene

An ethanolic extract was prepared from a crystalline lycopene preparation that was stored at −20 °C for about a year. 27.2 mg of this partially oxidized lycopene were extracted with ethanol and then evaporated under a vacuum, yielding 24 mg (~88% of the original lycopene). The extract was dissolved in 1.8 mL ethanol, and the resulting solution contained no detectable lycopene, as verified by measuring the absorption spectrum at 250–600 nm (not shown). The lycopene crystals that remained after the ethanol extraction (3.2 mg) were defined as intact lycopene based on the characteristic absorption spectrum (Figure 1b).

### 2.3. Synthetic Carotenoid Derivatives

Synthetic carotenoid derivatives, shown in Table 1 (>99% purity), were synthesized and provided by BASF (Dr. Hansgeorg Ernst, Ludwigshafen, Germany). The compounds, characterized using UV/VIS spectroscopy, HPLC, and 1H and 13C NMR, proved to be in an all-E-configuration.

### 2.4. Energy Calculations

The electronic structure method Restricted Hartree-Fock (RHF) was applied to resolve the chemical optimized structures [36] using the GAMESS suite of programs [37]. The basis set DZV was used to model all molecular orbitals. Atomic charges were computed using the Mulliken scheme, in which the atomic orbitals and molecular orbital coefficients were converted to an orthogonal set. These calculations provide electron populations that are less sensitive to basis set type [38]. In each molecule, there are two reactive carbon atoms in the conjugated chain. The Mulliken analysis was achieved for the two reactive carbon atoms (fourth position from both sides of the molecule, Table 1).

### 2.5. Solubilization of the Test Compounds

The synthetic derivatives were dissolved at 2 mM in THF and stored at −20°C. Before experiments, the absorption spectra of the compounds were checked for stability. Spectrophotometric analysis was performed at 250–600 nm using the V 530 UV/VIS spectrophotometer (Jasco, Easton, MD, USA). The THF stock solutions of each derivative were diluted in chloroform, and the concentrations were calculated according to the absorption values at the characteristic peaks [24]. The concentration of carotenoid solutions in the THF were calculated from the absorption after dilution in n-hexane: dichlomethane (5:1) containing 1.2 mM BHT.

Stock solutions were added to the cell culture medium under vigorous stirring and nitrogen flow to prevent oxidation. The final concentration of the carotenoids in the medium was measured by spectrophotometry after extraction in 2-propanol and n-hexane-dichloromethane. Stock solutions of curcumin (10 mM) were prepared in DMSO. Carnosic acid (30 mM) was dissolved in absolute ethanol. All procedures were done under reduced lighting, and the final concentrations of THF, ethanol, and DMSO in the medium were 0.75%, 0.15%, and 0.2%, respectively. The vehicles had no effect on the measured parameters.

### 2.6. Cell Culture

RAW264.7, murine monocyte-macrophage-like cells purchased from American Type Culture Collection (Manassas, VA, USA), were kindly provided by Dr. Bennie Gaiger (Weizmann Institute of Science, Rehovot, Israel). Cells were grown in DMEM (Gibco) with penicillin (500 units/mL), streptomycin (0.5 mg/mL), and 10% FCS. In all experiments, the cells were grown in α-MEM medium containing the same supplements, as well as RANKL (R&D systems, Minneapolis, MN, USA). Cells were grown in a humidified atmosphere of 95% air and 5% CO_2_, at 37 °C.

### 2.7. Differentiation Assays

Cells were seeded in 96-well plates (40,000 cells/mL), and test compounds were added 7–16 h later. In order to evaluate osteoclast differentiation, both TRAP activity and the number of TRAP-positive multinucleated cells were examined. TRAP activity in the cells was determined after 2–3 days by fixation with formaldehyde for 5 min (3.7% *v*/*v*) and washing with ethanol (95% *v*/*v*) for one min, followed by incubating the cells with 10–20 mM p-nitrophenyl phosphate (Sigma-Aldrich, St. Louis, MO, USA) in the presence of 10 mM sodium tartrate. The reaction was stopped with 0.1 M NaOH, and absorbance was measured at 410 nm. TRAP levels were corrected to cell number using crystal violet. Briefly, after fixation, cells were incubated for 15 min with crystal violet (0.5%) and washed thoroughly in tap water. After drying overnight, the dye was dissolved in sodium citrate, and absorbance was measured at 550 nm. It should be noted that the values of crystal violet staining, after treatment of cells with the various dietary compounds, did not differ by more than 20% from the value measured with RANKL alone. After 4 days, cells were fixed and stained for TRAP using a Leukocyte Acid Phosphatase kit (Sigma-Aldrich, St. Louis, MO, USA). The number of TRAP-positive multinucleated (>5 nucleus) cells was counted under a light microscope.

### 2.8. Cell Fractionation

Cells were seeded in 100-mm plates (3 × 10^6^ cells per plate). 16 h later, the test compounds were added for 3 h of pre-incubation. Then RANKL was added for 40 min. Cells were lysed with ice-cold cytosolic lysis buffer containing 10 mM NaCl, 10 mM Tris HCl (pH 7.4), 0.1 mM NP-40, 3 mM MgCl_2_, 1 mM EDTA, 2 mM sodium orthovanadate, 50 mM NaF, 0.2 mM DTT, and 1:25 Complete™ protease-inhibitor cocktail, and centrifuged at 310× *g* for 10 min at 4 °C. Supernatant samples were then centrifuged at 20,000× *g* for 10 min at 4 °C (cytosolic fraction). The pellet was resuspended with cytosolic lysis buffer and centrifuged (310× *g* for 10 min at 4 °C) twice. The nucleus pellet was lysed with nuclear lysis buffer containing 20 mM Hepes KOH (pH = 7.9), 1:4 glycerol, 420 mM NaCl, 1.5 mM MgCl_2_, 0.2 mM EDTA, 2 mM sodium orthovanadate, 50 mM NaF, 0.2 mM DTT, and 1:25 Complete™ protease-inhibitor cocktail, and incubated on ice for 20 min. The samples were centrifuged at 20,000× *g* for 10 min at 4 °C (nuclear fraction). Both fractions were further treated as described for Western blotting using the following antibodies: rabbit polyclonal IgG anti-p65 (#3034) (Cell Signaling Technology, Danvers, MA, USA), mouse monoclonal IgG anti-NF-κB p52 (sc-7386), goat polyclonal IgG anti-lamin B (sc-6216), rabbit polyclonal anti-b-tubulin (sc-9104) (Santa Cruz Biotechnology, Santa Cruz, CA, USA), and peroxidase-conjugated donkey anti-rabbit IgG (711-035-152) (Jackson Immunoresearch Laboratories, Inc. West Grove, PA, USA.).

### 2.9. Western Blotting

RAW264.7 cells were seeded in 100-mm plates (3 × 10^6^ cells per plate). 16 h later, the test compounds were added for 3 h of pre-incubation. Then RANKL was added for 15 min. Next, whole cell extracts were prepared. Briefly, cells were lysed in ice-cold lysis buffer containing 50 mM, HEPES (pH 7.5), 150 mM NaCl, 10% (*v*/*v*) glycerol, 1% (*v*/*v*) Triton X-100, 1.5 mM EGTA, 2 mM sodium orthovanadate, 20 mM sodium pyrophosphate, 50 mM NaF, 1 mM DTT, and 1:25 Complete™ protease-inhibitor cocktail (Roche Molecular Biochemicals, Mannheim, Germany), and centrifuged at 20,000× *g* for 10 min at 4 °C. 50 μg protein of the supernatants were separated by SDS-PAGE, and then blotted into nitrocellulose membrane (Whatman, Dassel, Germany). The membranes were blocked with 5% milk for 2 h and incubated with primary antibodies overnight at 4 °C, followed by incubation with peroxidase--conjugated secondary antibodies (Promega, Madison, WI, USA) for 2 h. The protein bands were visualized using Western Lightning™ Chemiluminescence Reagent Plus (PerkinElmer Life Sciences, Inc., Boston, MA, USA). The blots were stripped and re-probed for the constitutively present protein calreticulin, which served as the loading control. The optical density (OD) of each band was quantitated using ImageQuant TL7.0 (GE Healthcare, Chicago, IL, USA). The following antibodies were used: mouse monoclonal IgG anti-phospho-IκBα Ser32/36 (#9246) (cell signaling technology), mouse monoclonal IgG anti-IκB (OP142) (Oncogene Research Products, La Jolla, CA, USA), and rabbit polyclonal IgG anti-calreticulin (PA3-900) from Affinity BioReagent (Golden, CO, USA).

### 2.10. Statistical Analysis

All experiments were repeated at least three times. The significance of the differences between the means of the various subgroups was assessed by a two-tailed Student’s *t* test using Microsoft Excel. Statistically significant differences among the multiple groups were analyzed by a one-way ANOVA, followed by a Newman–Keuls multiple comparison test using the GraphPad Prizm 5.0 program (GraphPad Software, San Diego, CA, USA). *p* < 0.05 was considered statistically significant. The interaction between the polyphenols and the carotenoid derivatives in inhibiting TRAP activity was assessed by CI analysis using Calcusyn version 2.1, (BIOSOFT, Cambridge, Great Britain). The CI values were calculated based on the % inhibition by each agent individually and by the combinations at a constant ratio. 

## 3. Results

### 3.1. Oxidized Lycopene Is More Potent than Intact Lycopene in Inhibiting RANKL-Induced Osteoclast Differentiation

Using a partially oxidized lycopene, we separated the hydrophobic intact lycopene, which is not soluble in ethanol, from its more hydrophilic oxidation products by ethanol extraction. The hydrophilic fraction comprised about 89% by weight of the oxidized lycopene preparation. The spectral absorption of the non-extracted oxidized lycopene preparation (Figure 1a) showed higher absorption in the 300–400 nm range than that of the intact lycopene preparation (Figure 1b), suggesting that the latter does not contain a considerable amount of oxidized derivatives. We examined the effect of these intact and oxidized preparations of lycopene in the inhibition of RANKL-induced osteoclast differentiation in RAW264.7 cells. The picture in Figure 1c shows small monocytes in the control and large multinucleated osteoclasts in the RANKL-treated cells. Similar osteoclasts are seen in cells treated with RANKL and intact lycopene, in contrast to cells treated with RANKL and oxidized lycopene that showed no multinucleated osteoclasts, suggesting that the oxidized lycopene inhibited osteoclast differentiation. A quantitative analysis showed that the oxidized lycopene preparation was much more potent in inhibiting TRAP activity and formation of TRAP+ multinucleated osteoclasts than the intact lycopene (Figure 1d). To evaluate whether the treatment of cells with oxidized or intact lycopene affect cell survival, the values of cell protein, measured by crystal violet staining (used to normalize TRAP results), was compared to that of cells treated with RANKL alone. The average of four experiments, each performed in triplicate was 46,700 ± 2000, 44,600 ± 4300, and 49,100 ± 5400 for RANKL alone, RANKL with oxidized lycopene, and RANKL with intact lycopene, respectively. Thus, the results presented in Figure 1d represent inhibition of osteoclast differentiation and are not attributed to cell death. 

### 3.2. Diapocarotene-Dials Inhibition of RANKL-Induced Osteoclast Differentiation Depends on the Electron Density around the Reactive Carbon Atoms of the Molecules

To determine the effect of diapocarotene-dials on RANKL-induced osteoclast differentiation in RAW264.7 cells, we incubated these cells with RANKL and with different concentrations of 6,14′-diapocarotene-6,14′-dial (6,14′), and assessed TRAP activity and the formation of multinucleated TRAP+ cells. The percent inhibition by 6,14′ was similar for the two measured parameters. The inhibition of osteoclast differentiation by this derivative was dose dependent, and almost complete inhibition was observed at 10 µM (Figure 2a). We measured TRAP activity and the formation of multinucleated TRAP+ cells with 10 µM of two different diapocarotene-dials (6,14′ and 10,10′). The percent inhibition by each compound was similar for the two measured parameters, and 6,14′ was more active than 10,10′ (Figure 2b). Treatment of cells with diapocarotene-dials alone without RANKL did not result in any response (data not shown). Different diapocarotene-dials inhibited osteoclast differentiation to different extents (Figure 2c). In previous work, we have shown that the activity of individual carotenoid derivatives in inhibiting the NF-κB reporter gene activity [25] depends on the electron density around the reactive carbon atoms (the fourth atom from each side of the molecule). Since NF-κB is known to partially mediate RANKL signaling, we assumed that RANKL-induced osteoclast differentiation would similarly depend on the structure of the diapocarotene-dials. Indeed, a strong correlation (R^2^ = 0.938) exists between the electron density at the reactive C-atom of the various diapocarotene-dials (Table 1) and the % inhibition of TRAP activity (Figure 2c). The results strengthen the evidence that the potency of these derivatives depends on the electron density around the reactive carbon atoms. 

### 3.3. Diapocarotene-Dials Inhibit RANKL-Induced NF-κB Activation in Osteoclast Precursors

Activation of NF-κB is comprised of two pathways: the canonical and the alternative or non-canonical. Phosphorylation and degradation of its inhibitory subunit IκBα is an essential step in activating the canonical pathway. Western blot analysis revealed that the active diapocarotene-dials 6,14′ and 10,10′ significantly inhibit RANKL-induced IκBα phosphorylation and degradation, as opposed to the inactive diapocarotene-dial 8,8′ and 12,12′ (Figure 3a,b), in accordance with the structure-activity relationship described above. It is noticeable in Figure 3a that the level of pIκB in the 6,14′-treated sample was greater than with RANKL alone; however, quantitating the pIκB:IκB ratio (Figure 3b, corrected to calreticulin) clearly shows that 6,14′ treatment reduced this ratio by more than 30%, which indicates downregulation of IκBα phosphorylation and inhibition of RANKL-induced degradation of IκBα. IκBα degradation enables the translocation of p65 to the nucleus. Fractionation analysis of nuclear p65 (Figure 3c,d) shows some reduction in the nucleus after treatment with lycopene or its active derivatives; however, this reduction was not statistically significant, but may suggest that the active derivatives attenuate the canonical pathway of NF-κB. 

Degradation of the precursor p100 to the active NF-κB component p52 is essential in activating the alternative pathway. RANKL reduced the nuclear level of p100 and increased that of p52 (Figure 3e–g). Treatment with lycopene, 6,14′ and 10,10′ suggests inhibition of the RANKL-induced conversion of p100 to p52, but the changes were not significant. In addition, these compounds preserved the cytosolic levels of p52 and prevented its translocation to the nucleus (Figure 3e). These results may suggest that active diapocarotene-dials inhibit both pathways in RANKL-induced NF-κB activation in RAW264.7 cells.

### 3.4. Active Diapocarotene-Dials Inhibit RANKL-Induced TRAP Activity Synergistically with Curcumin and with Carnosic Acid 

RAW264.7 cells were incubated with combinations of the diapocarotene-dial 6,14′, with the polyphenols curcumin and carnosic acid. At low concentrations of each agent, these combinations produced a synergistic anti-differentiative effect in RANKL-induced cells. Synergistic effects were evaluated using Calcusyn Software for Dose Effect Analysis. Dose effect curves and CI values for the combination of 6,14′ with curcumin (Figure 4a,b) and 6,14′ with carnosic acid (Figure 4c,d) are presented. Most CI values are below 1.0, indicating some synergy at most of the tested concentrations. However, CI values at low concentrations, resulting in 20–40% inhibition, are smaller than 0.5, indicating a strong synergistic effect at concentrations that can be found in human blood.

## 4. Discussion

The main finding of the current study is that the inhibition of RANKL-induced osteoclast differentiation by partially oxidized lycopene is mediated by the hydrophilic oxidation products, and not by the intact lycopene molecule. Two different approaches led us to this conclusion. (a) We separated the spontaneously oxidized derivatives from the intact carotenoid using an ethanolic extraction of a partially oxidized lycopene preparation, and found that the oxidized lycopene inhibited osteoclast differentiation, whereas the parent molecule was nearly ineffective. (b) Using a series of fully characterized synthetic diapocarotene-dials, we found that these compounds inhibited osteoclast differentiation, and the inhibition efficiency correlated with the reactivity of the α,β-unsaturated carbonyl groups in reactions such as Michael addition. This reactivity was estimated by calculating the electron density around the reactive carbon atoms, as shown in Table 1. The relative effectiveness of the diapocarotene-dials in the inhibition of osteoclastogenesis was similar to that found for activation of the ARE/Nrf2 transcription system [24] and for inhibition of TNFα-induced NFkB activity [25] by carotenoid derivatives. Thus, it is suggested that the inhibition of RANKL-induced osteoclast differentiation resulted, at least partially, from inhibition of the RANKL-activated NF-κB activity. In support of this suggestion, we found that the effective diapocarotene-dials reduced activation of the canonical NFκB pathway by RANKL. This was evident from the reduction of IκBα phosphorylation and degradation, and of p65 nuclear translocation that are essential stages in the canonical pathway. The active diapocarotene-dials and lycopene also reduced the nuclear translocation of p52, suggesting inhibition of the non-canonical NFκB pathway that is known to be involved in RANKL-induced osteoclast differentiation [17,20].

Several proteins that take part in the NFκB pathway (e.g., IκB kinase) and the NFκB subunits (e.g., p65) contain cysteine residues which regulate NFκB activity [39,40]. The interaction of electrophiles with these cysteine thiols leads to NFκB pathway inhibition [39]. Similarly, it was shown that sulforaphane, its analogs [41], and other electrophiles such as carnosic acid [42], interact with reactive cysteine thiols in the Keap1 protein, leading to activation of the ARE/Nrf2 transcription system. Hydrophobic carotenoids such as lycopene and beta carotene are devoid of electrophilic groups which can interact with these cysteines; however, we previously demonstrated that apo-carotenal derivatives interact directly with thiol groups of IκB kinase [25]. In addition, we previously suggested, although did not directly prove, that carotenoid-oxidized derivatives activate the Nrf2 transcription system by interaction with such reactive cysteines in the Keap1 protein [24]. Since NFκB is involved in RANKL activation of osteoclast differentiation, and reduction of RANKL-induced ROS generation through activation of ARE/Nrf2 was suggested to inhibit this differentiation [35], we propose that the interaction of carotenoid derivatives with thiol groups in proteins critically involved in NFκB and ARE/Nrf2 pathways may be part of the mechanism for the inhibition of osteoclast differentiation by oxidized derivatives of lycopene and other carotenoids.

As all the effects of carotenoid derivatives were obtained in in-vitro cellular systems, an important question is whether such effects can also be obtained in-vivo. Although this question is difficult to answer directly, what we can try to resolve is whether such apo-carotenals can be found in mammalian blood and tissues, and what their potential sources are. It is possible that the derivatives are consumed with the carotenoids from foods or formed inside the body. Diapocarotene-dials similar to those used in the current study were only rarely found in plants, most likely because they are reactive and instable molecules, which makes them difficult to detect in biological samples [43]. Recently, Jia et al. identified in Arabidopsis a presumed carotenoid-derived dialdehyde, anchorene, (12,12′-diapocaroten-12,12′-dial according to our nomenclature) that promotes the development of anchor roots [44]. However, such rare plant metabolites probably have no importance in the human diet and, thus, it is not surprising that diapocarotene-dials have not been detected in mammalian samples. In contrast, mono-apocarotenals, both, β-apo-carotenals [45] and lycopenals [46], were identified in both human and plant samples. Specifically, apolycopenals including apo-10′-, apo-12′-, apo-14′-, and apo-15′-lycopenal were found in foods that are rich in lycopene such as raw tomatoes, red grapefruit, and watermelon. These lycopenals were also detected in the plasma of individuals who had consumed tomato juice for 8 weeks [46]. Similar compounds, apo-8′- and apo-12′-lycopenals, were detected in rat livers [47], and in a recent study, additional apo-carotenals were detected, including β-apo-12′-carotenal and several apo-zeaxanthinals and apo-luteinals [48]. The concentration of these apo-lycopenals in foods is very low, and is about 500 times lower than that of lycopene [49], with similar relative concentrations found in human plasma. However, it is not clear if apo-carotenals are absorbed from foods or produced in the body since it was found that 4-week supplementation of high-β-carotene and high-lycopene tomato juice did not lead to detectable concentrations of most β-apocarotenals or lycopenals that were present in the juice [50]. It is not certain if the accessibility of these compounds to bone and other cells in-vivo is sufficient to achieve the beneficial effects. Another alternative is that the apo-carotenals are formed inside the cells from the intact carotenoids, close to the site of their activity. Carotenoids are cleaved in the cells by the central cleavage enzyme 15,15′-β-carotene oxygenase 1 (BCO1) and by the eccentric cleavage enzyme β,β-carotene-9′,10′-oxygenase 2 (BCO2). The latter enzyme exhibits broad substrate specificity and cleaves both carotenes, such as lycopene, and xanthophylls like lutein [51]. The cleavage at the 9,10 double bond results in the formation of apo-10′-carotenals and 10,10′-diapocarotenals [52]. The 10,10′-diapocaroten-10,10′-dial was formed in vitro by incubation of β-carotene, as well as other carotenoids, with a recombinant BCO2 [53], but they were not detected in mammalian samples perhaps because of their high reactivity in biological systems [24]. Although, in the current study, we analyzed the activity of only the diapocarotenal, the reactivity of the apo-10′-lycopenals in the activation of the ARE/Nrf2 transcription system was only 1.5–2.5 fold lower than that of the 10,10′-diapocaroten-10,10′-dial [24]. Thus, it is possible that formation inside the osteoclasts may result in high enough local concentrations to lead to the inhibition of osteoclast differentiation. 

Similar to carotenoid derivatives, other phyto-nutrients are known to inhibit osteoclast differentiation. These include flavonoids such as quercetin [54], polyphenols such as curcumin [55], and resveratrol [56], sulforaphane [57], and other isothiocyanates [58]. Although these nutrients have different chemical structures, they, and the carotenoid derivatives, all have electrophilic groups in common that can interact with thiol groups of reactive proteins of the NFκB system [39,40] or other signaling pathways involved in osteoclast differentiation. A significant inhibition of osteoclastogenesis by the two polyphenols and by the carotenoid derivatives tested in the current study occur at high concentrations (6,14′ and curcumin—above 2 µM; carnosic acid—above 10 µM). Since, usually, these concentrations cannot be achieved in-vivo, we tested if their combination would result in activity at concentrations that could be achieved. We found a strong synergy between the polyphenols and the carotenoid derivative, 6,14′, leading to significant inhibition at concentrations below 1 µM. However, understanding the mechanism of this synergy would require extensive research to explore whether it results from synergistic inhibition of NFκB at different elements of the pathways or from interference in other pathways leading to osteoclast differentiation. For example, curcumin has been shown to inhibit the differentiation of human monocytes to osteoclasts by reducing phosphorylation and activation of mitogen-activated protein kinase (MAPK) proteins, such as ERK, p38, and JNK, which leads to reduced expression of c-Fos and NFATc1 that are essential for differentiation of osteoclasts [59]. Similar reduction of the phosphorylation of ERK, p38, and JNK MAPKs by carnosic acid was evident in RANKL-induced RAW264.7 cells, followed by a decrease in expression of c-Fos and NFATc1 and inhibition of osteoclastogenesis [35]. Thus, inhibition of MAPKs by the polyphenols, curcumin and carnosic acid, can increase the inhibitory effect of carotenoid derivatives that reduce RANKL-induced NFκB activation. Thummuri, et al. have shown in RAW264.7 cells and in mouse bone marrow macrophages that activation of ARE/Nrf2 and reduction of RANKL-induced ROS generation is one of the mechanisms for carnosic acid inhibition of osteoclastogenesis [35]. This is another possible explanation for the synergy we presented between the polyphenols and the carotenoid derivatives, as we have recently shown a strong synergy in ARE/Nrf2 activation in human keratinocytes by combinations of lycopene or tomato extract with carnosic acid or curcumin [60].

## 5. Conclusions

The current paper suggests that the protective effect of lycopene and other carotenoids on bone health, as shown in population and animal studies, is at least partially related to the inhibition of osteoclast differentiation and activity. This inhibition is possibly associated with that of the NFκB transcriptional system. Although most previous studies were done with carotenoids in foods or with pure carotenoids, we suggest that the osteoclasts are actually affected by the apo-carotenal carotenoid derivatives and not by the intact molecules.

## Figures and Tables

**Figure 1 antioxidants-09-01167-f001:**
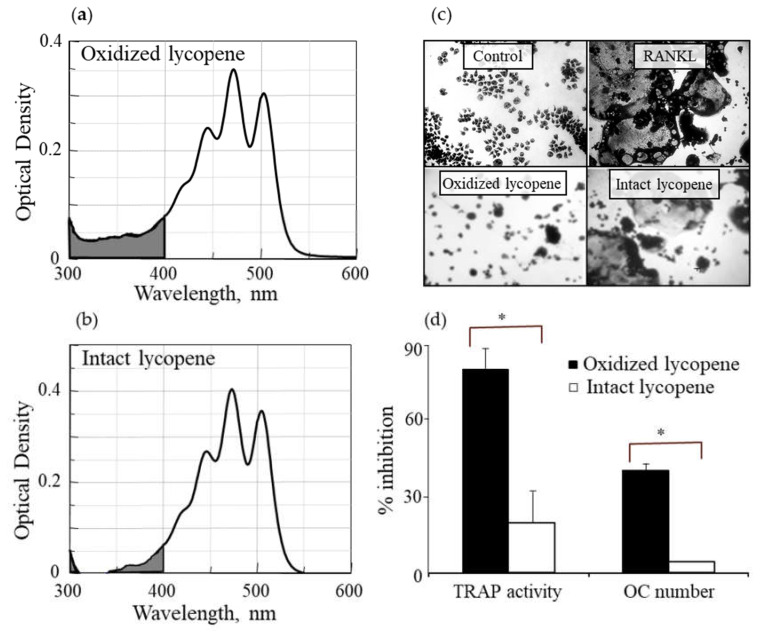
Oxidized lycopene inhibits osteoclast differentiation. Characteristic absorption spectrum of the oxidized lycopene (**a**) and intact lycopene (**b**) used in the experiment. (**c**,**d**) RAW264.7 cells (4 × 10^4^ cells/well) were incubated either alone or in the presence of RANKL (20 ng/mL) without lycopene or with one of its two types at a concentration of 10 µM. (**c**) Photographs of cells after staining for tartrate resistant acid phosphatase- (TRAP)-positive cells (original magnification × 100). (**d**) Counting of multinucleated TRAP-positive cells and measurement of TRAP activity. Values are the means ± SD of three experiments, each performed in triplicate. * *p* < 0.01 for the difference between the % inhibition with oxidized lycopene vs. intact lycopene.

**Figure 2 antioxidants-09-01167-f002:**
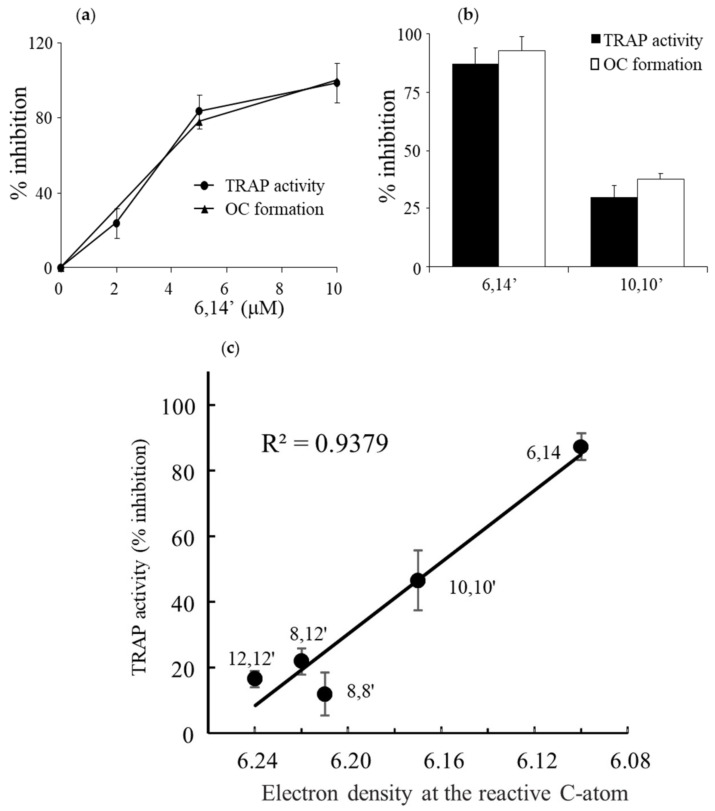
Diapocarotene-dials inhibit RANKL-induced osteoclastogenesis. Osteoclast differentiation was measured as described in the Materials and Methods section and in Figure 1. Cell were incubated with RANKL alone or with (**a**) different concentrations of the diapocarotene-dial 6,14′ or (**b**,**c**) with 10 µM different diapocarete-dials. Inhibition is shown in relation to positive control with RANKL. (**b**) Comparison of the inhibition by 6,14′ and 10,10′. Values are the means ± SE of 3–14 experiments, each performed in triplicate, *p* < 0.01 for the difference between the % inhibition with 6,14′ vs. 10,10′. (**c**) Correlation between the electron density at the reactive C-atom of the various diapocarotene-dials and the % inhibition of osteoclast TRAP activity. Values are the means ± SE of 3–11 independent experiments, each performed in triplicate. The results are statistically significant (ANOVA test) *p* < 0.05.

**Figure 3 antioxidants-09-01167-f003:**
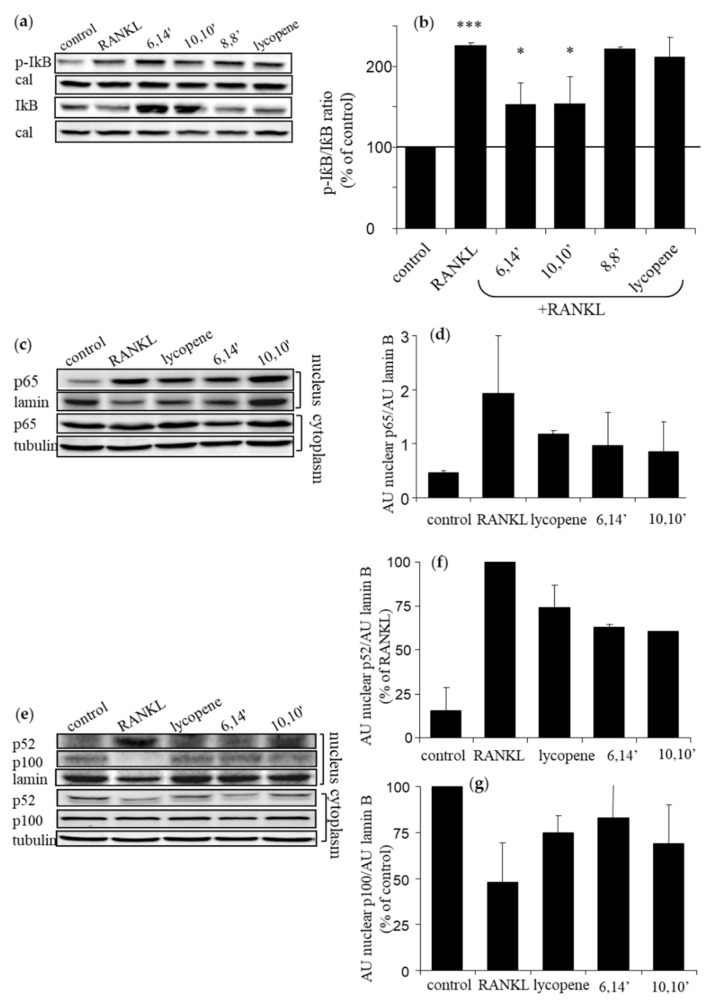
Diapocarotene-dials attenuate the NF-κB signal in RANKL-activated RAW264.7 cells. 5 × 10^6^ cells in 100-mm plates were either incubated alone or in the presence of the indicated diapocarotene-dials (10 µM) or lycopene (10 µM) for 2 h, and then treated with RANKL (40 ng/mL) for 15 min (**a**,**b**) and 40 min (**c**–**g**). Whole cell lysates (**a**,**b**) and cytoplasmatic and nuclear fractions (**c**–**g**) were prepared and analyzed by Western blotting, as described in Materials and Methods. Values are the means ± SE of three independent experiments. (**a**) Blots of IκBα and pIκB. (**b**) The ratio of p-IκB:IκB after normalization with calreticulin (cal) is presented as the % of the control without RANKL. *** *p* < 0.001 for the difference between RANKL to the control. * *p* < 0.05 for the difference between RANKL to 6,14′ and 10,10′. (**c**) Blots of nuclear and cytosolic p65. (**d**) Nuclear p65 levels normalized to laminin B. (**e**) Blots of nuclear and cytosolic p52 and p100. (**f**) Nuclear p52 levels normalized to laminin B. Results are % of RANKL. (**g**) Nuclear p100 levels normalized to laminin B. Results are % of control.

**Figure 4 antioxidants-09-01167-f004:**
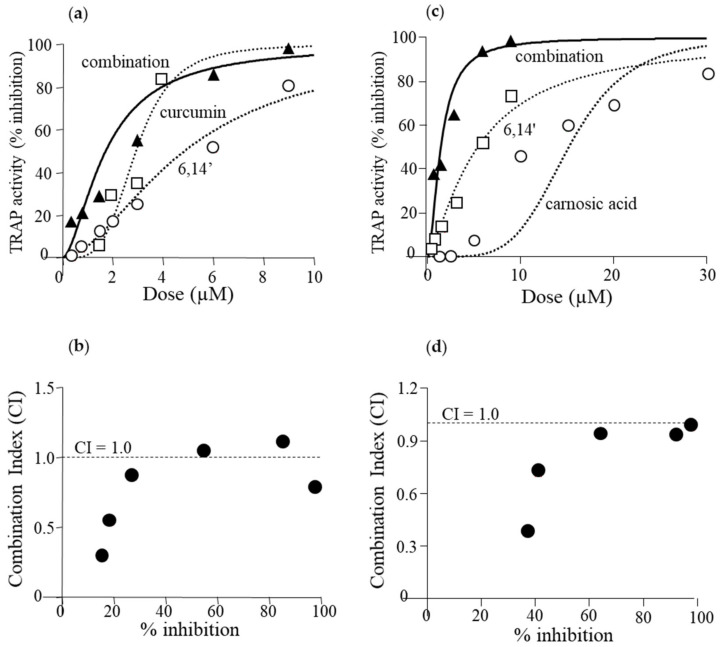
Synergistic effect of 6,14′ with curcumin or carnosic acid. Osteoclast differentiation was measured as described in the Materials and Methods section and in Figure 1. Cell were incubated with RANKL alone or in the presence of different concentrations of 6,14′ with curcumin or with carnosic acid at constant concentration ratios. Values are the means of 3–4 experiments, each performed in triplicate. (**a**) The dose effect curve of the combinations of 6,14′ with curcumin at a ratio of 1:1. (**b**) Combination index (CI) values for the combinations of 6,14′ with curcumin. (**c**) The dose effect curve of the combinations of 6,14′ with carnosic acid at a ratio of 1:3. (**d**) CI values for the combinations of 6,14′ with carnosic acid.

**Table 1 antioxidants-09-01167-t001:** Structures, Mulliken population values, and HOMO-LUMO energy gap of the synthetic derivatives.

Derivative ^1^	Structure	Mulliken Population Values (Electron Density)	HOMO-LUMO ^2^ Energy Gap (kcal/mol)
		Left	Right
6,14′	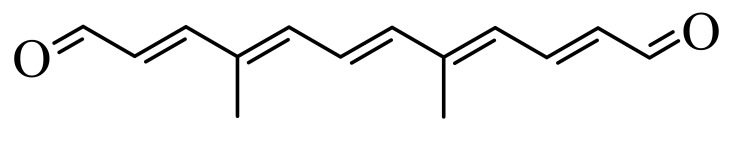	6.16	6.10	189.51
10,10′	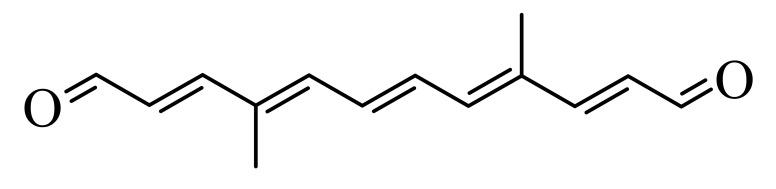	6.17	6.17	191.39
8,8′	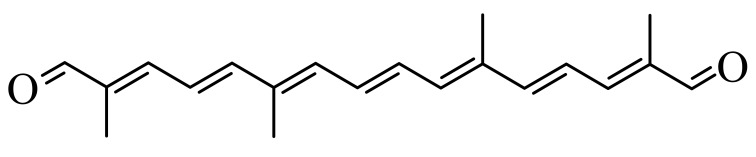	6.21	6.21	178.21
8,12′	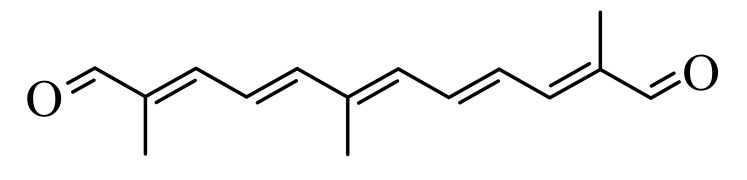	6.23	6.22	210.84
12,12′	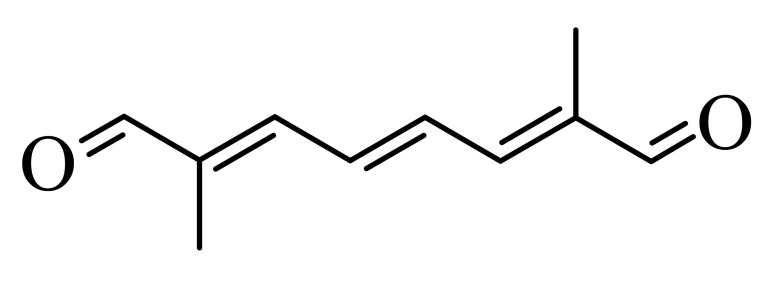	6.24	6.24	214.61

^1^ The abbreviated names of the derivatives are derived from the putative position of oxidative cleavage in the carotenoid backbone, which could lead to the formation of these derivatives, Full names: 6,14′-diapocarotene-6,14′-dial (6,14′); 10,10′-diapocarotene-10,10′-dial (10,10′); 8,8′-diapocarotene-8,8′-dial (8,8′); 8,12′-diapocarotene-8,12′-dial (8,12′); 12,12′-diapocarotene-12,12′-dial (12,12′). ^2^ HOMO: High Occupied Molecular Orbitals; LUMO: Low Unoccupied Molecular Orbitals.

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
