# Peer review of "Inhibition of Osteoclast Differentiation by Carotenoid Derivatives through Inhibition of the NF-κB Pathway"

_antioxidants, 2020, doi:10.3390/antiox9111167_

Round 1

Reviewer 1 Report

In the manuscript, it was suggested too loosely that the study showed a synergistic inhibition of osteoclast differentiation by combination of lycopene oxidation products and other anti-oxidative materials (already well known, curcumin and carnosic acid). It seems to need more study to clarify a mechanism for the synergistic inhibition of osteoclast differentiation by combination of lycopene oxidation products and curcumin/carnosic acid.

Please revise the manuscript with followings.

  1. correct "CO2" in line 165.
  2. correct "106" in line 180 and line 196.
  3. check "MgCI2" in line 182.
  4. check "phospho-I?B?" in line 209.
  5. check "anti-I?B?" in line 210.
  6. which of "RAW 264.7 (in line 196) and RAW264.7(in line 234) " is correct? It needs to be unified.
  7. check "Student's t" with "Student's t (italics)".
  8. which of "P<0.01 (in line 218) and p<0.05 (in line 283) " is correct? It needs to be unified.
  9. check "subunit I?Ba" in line 286.
  10. correct "104 cells" in line 328.
  11. check "obtained in vivo" in line 370.

Author Response

The reviewer asks for "more study to clarify a mechanism for the synergistic inhibition". We did not ignore this important issue and discussed the possible mechanisms for the synergy toward the end of the manuscript. However, we believe that such additional studies are beyond the scope of the current manuscript and would require a long time to evaluate even part of the mechanisms suggested in the discussion. Because Reviewer 3 suggested to clarify this point, more information was added to this discussion.

Minor errors indicated in 1–10 were corrected. 6. RAW264.7 was used throughout the manuscript. 8. p < 0.0X was used throughout the manuscript. Comment 11 is not clear.

Reviewer 2 Report

The authors showed that  lycopene  oxidation  product, diapocarotene-dials, is more  potent  than  intact  lycopene  in  inhibiting  osteoclast  differentiation. As mechanism,  the  carotenoid  derivatives  attenuated  the  NF-ƙB  signal  through  inhibition  of IƙB  phosphorylation  and  NF-ƙB  translocation  to  the  nucleus. Overall, the manuscript is well-written and the study provided interesting insight into the function of anti-oxidants in osteoclast. There are some concerns that authors should address.

Does oxidized lycopene affect cell proliferation or cell survival?

Fig3 (d)-(g) Are they statistically different?

Author Response

Does oxidized lycopene affect cell proliferation or cell survival?

This is an important issue and, thus, we added information about the similar number of cells after treatment with RANKL alone or with oxidized or intact lycopene (see lines 243-248). A conclusion that cell survival was not affected by these compounds was also added (lines 248-249).

Fig3 (d)-(g) Are they statistically different?

The reviewer is right that in Figure 3d, f, and g, the results are not statistically different, and no p values were presented for these results. To clarify this point, we added this information to the text and changed the description of the results to conform with the lack of statistical difference and to only suggest that lycopene and its derivatives inhibit both pathways of NF-ƙB activation (see lines 302-304).

Reviewer 3 Report

The manuscript numbered 980465ISSN with title “Inhibition of osteoclast differentiation by carotenoid derivatives through inhibition of the NF-ƙB pathway” is an interesting work concerning the protective effect of lycopene and other carotenoids on bone health. However, in my opinion, the manuscript is recommended for publishing after minor revisions.

Same points of the manuscript should be improved.

  1. The grammatical English language should be improved and clarified.
  2. The introduction section is supported by existing literature in the field highlighting the relevance of this research. However, this section should be better-organized underlying objectives of the work.
  3. In the paper, it was evaluated the effect of lycopene intact and oxidation products in the inhibition of RANKL-induced osteoclast differentiation by using murine monocyte-macrophage-like cells RAW264.7. The osteoclast differentiation inhibition was explained by the interaction of lycopene with reactive cysteines of NFkB subunits. This interaction should be better justified.
  4. The figures captions are too long. The description of the captions should be reduced.
  5. The authors show that the combination of two polyphenols such as carnosic acid and curcumin with carotenoid derivates such lycopene lead to a synergistic effect on inhibition of osteoclastogenesis. The mechanism responsable of this synergic activity should be better clarified.

Author Response

The numbers refer to the numbers in the reviewer's comments

The indicated line numbers are in the clean, unmarked revised manuscript

  1. The English language of the manuscript was edited by a professional native English-speaking editor, and we have never encountered any problems in other journals. However, if the journal editors believe that the language should be improved, you can suggest an appropriate editorial service, and we will pay for it.
  2. We understand this comment as a request to better organize the work's objective. Thus, we expanded this part to clarify the aims and link them to the introduction. (see lines 90-92, 93–98, 100-102).
  3. This issue was explained to some extent in the original manuscript. We now expanded the discussion to better clarify the possible interaction of lycopene derivatives with thiol groups (see many changes in lines 365-379).
  4. Most figures captions were reduced, and some of the information was transferred to the Materials and Methods section. We believe that further reduction would prevent proper evaluation of the results.
  5. We shortly discussed, in the original manuscript, possible mechanisms for the synergy. More details were added to this discussion in order to clarify it, and the suggested mechanisms were more clearly defined.

Round 2

Reviewer 2 Report

The authors have addressed all my comments. The current manuscript is suitable for publication.